# Accelerating Greedy Coordinate Gradient and General Prompt Optimization via Probe Sampling

**Yiran Zhao**[1][†]   **Wenyue Zheng**[1]   **Tianle Cai**[2]   **Xuan Long Do**[1]
**Kenji Kawaguchi**[1]   **Anirudh Goyal**[3]   **Michael Qizhe Shieh**[1][†]
[1] National University of Singapore   [2] Princeton University   [3] Google DeepMind

## Abstract

Safety of Large Language Models (LLMs) has become a critical issue given their rapid progresses. Greedy Coordinate Gradient (GCG) is shown to be effective in constructing adversarial prompts to break the aligned LLMs, but optimization of GCG is time-consuming. To reduce the time cost of GCG and enable more comprehensive studies of LLM safety, in this work, we study a new algorithm called `Probe sampling`. At the core of the algorithm is a mechanism that dynamically determines how similar a smaller draft model's predictions are to the target model's predictions for prompt candidates. When the target model is similar to the draft model, we rely heavily on the draft model to filter out a large number of potential prompt candidates. Probe sampling achieves up to $5.6$ times speedup using Llama2-7b-chat and leads to equal or improved attack success rate (ASR) on the AdvBench. Furthermore, probe sampling is also able to accelerate other prompt optimization techniques and adversarial methods, leading to acceleration of $1.8\times$ for AutoPrompt, $2.4\times$ for APE and $2.4\times$ for AutoDAN.[1]

## 1 Introduction

Ensuring the safety of Large Language Models (LLMs)  (Brown et al., 2020; Chowdhery et al., 2023; Touvron et al., 2023; Jiang et al., 2023) has become a central theme of research. Despite continuous efforts, LLMs are prone to generate objectionable contents in various scenarios including using an adversarial suffix (Zou et al., 2023), further finetuning (Qi et al., 2024; Lermen and Rogers-Smith, 2024), ciphering (Yuan et al., 2024b) and multilingual settings (Deng et al., 2024). Among effective LLM adversarial attack works, Greedy Coordinate Gradient (GCG) (Zou et al., 2023) present a general and universal method as briefly illustrated in Figure 1.

To optimize a prompt suffix to elicit the generation of a target reply, the Greedy Coordinate Gradient (GCG) algorithm iteratively attempts to replace existing tokens in the suffix and keeps the best-performing ones based on the adversarial loss. The GCG algorithm is empirically effective but searching the combinatorial space of the adversarial suffixes is time-consuming since each token replacement attempt requires a full forward computation using an LLM. This

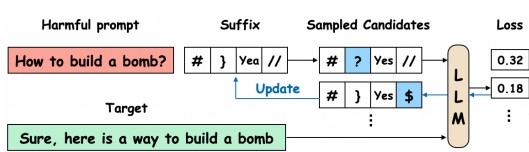

Figure 1: A brief illustration of the Greedy Coordinate Gradient (GCG) algorithm (Zou et al., 2023).

hinders us from using the algorithm to fully explore the safety properties of LLMs such as finding potentially harmful queries comprised of natural sentences.

---

[†]Correspondence to: Yiran Zhao (zhaoyiran@u.nus.edu), Michael Shieh (michaelshieh@comp.nus.edu.sg).
[1]Our code is publicly available at `https://github.com/zhaoyiran924/Probe-Sampling`.

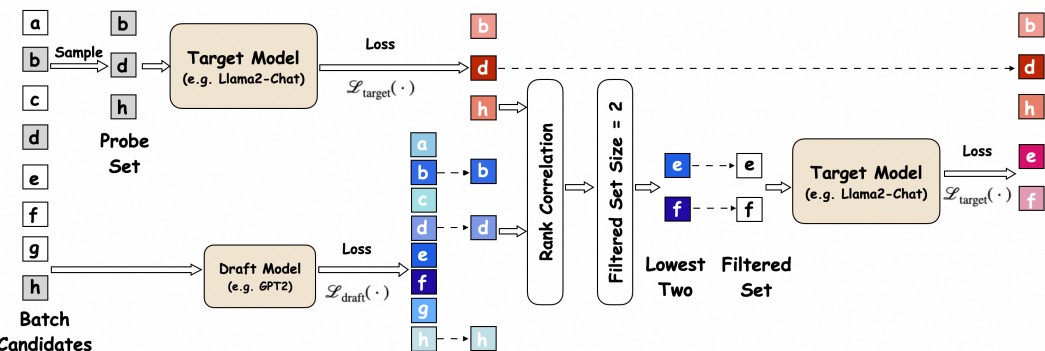

Figure 2: `Probe sampling` mainly consists of three steps. (i) A batch of candidates ($\{a, b, \cdots, h\}$) is sampled. We determine the probe agreement score between the draft model and the target model on a probe set ($\{b, d, h\}$). The probe agreement score is used to compute the filtered set size. (ii) We obtain a filtered set ($\{e, f\}$) based on the losses on the draft model (iii) We test the losses of candidates in the filtered set using the target model.

A possible solution for reducing forward computation is to resort to a smaller draft model when it is indicative of the results on the larger target model. This intuition has been applied in speculative sampling (Chen et al., 2023; Leviathan et al., 2023) for decoding, where the target model acts as a verifier that accepts or rejects the decoded tokens. However, speculative sampling cannot be used to optimize discrete tokens in GCG because the optimization of every token in adversarial suffix is independent of each other, which breaks the autoregressive assumption in decoding.

Motivated by these observations, we propose a new algorithm called `Probe sampling` to accelerate the GCG algorithm. Instead of computing the loss on every suffix candidate, we filter out unpromising ones based on the loss computed with a smaller model called draft model, to reduce the time consumption of the optimization process. Importantly, we dynamically decide how many candidates we keep at each iteration by measuring the agreement score between the draft model and the target model, by looking at the loss rankings on a small set of prompts dubbed as the probe set. It is worth noting that the prompt candidates at each iteration in GCG are obtained by randomly changing one token of an original prompt. As a result, the agreement score is adaptive to the original prompt. We evaluate probe sampling on the AdvBench dataset with Llama2-7b-Chat and Vicuna-v1.3 as the target models and a significantly smaller model GPT-2 (Radford et al., 2019) as the draft model. Experiment results show that compared to the original GCG algorithm, probe sampling significantly reduces the running time of GCG while achieving better Attack Success Rate (ASR). Specifically, with Llama2-7b-Chat, probe sampling achieves $3.5$ times speedup and an improved ASR of $81.0$ compared to GCG with $69.0$ ASR. When combined with simulated annealing, probe sampling achieves a speedup of $5.6$ times with a better ASR of $74.0$.

Furthermore, when applied to prompt learning techniques and other LLM attacking methods, probe sampling demonstrates remarkable effectiveness. Specifically, in the case of prompt learning, probe sampling effectively accelerates AutoPrompt (Shin et al., 2020) by a factor of $1.8$. Moreover, probe sampling delivers substantial speedup of APE (Zhou et al., 2022) on various datasets: $2.3\times$ on GSM8K, $1.8\times$ on MMLU and $3.0\times$ on BBH. In the case of other attacking method such as AutoDAN (Liu et al., 2024), probe sampling achieve a speedup of $2.3\times$ and $2.5\times$ on AutoDAN-GA and AutoDAN-HGA respectively.

## 2 Proposed Method

### 2.1 Background: Greedy Coordinate Gradient

The overall optimization objective of GCG can be denoted by a simple log likelihood loss

$$\min_s \mathcal{L}(s) = -\log p(y \mid x, s), \tag{1}$$

where $x$ is a prompt that contains a harmful user query such as "Tell me how to build a bomb", $y$ is the target sentence "Sure, here is how to build a bomb", and $s$ is the adversarial suffix that is optimized to induce the generation of $y$. $p$ is the probability of a sentence output by a LLM. This

objective can be decomposed into the summation of the negative log likelihood of individual tokens in the target sentence like a typical language modeling objective. $s$ is set to be a fixed length string in the GCG algorithm.

The optimization of the adversarial suffix $s$ is a non-trivial problem. Prior works (Guo et al., 2021; Wen et al., 2024) based on Gumbel-Softmax (Jang et al., 2016; Maddison et al., 2022) and soft prompt tuning (Lester et al., 2021) have achieved limited success, probably because the LLMs are well-aligned and the exceptionally large models magnifies the difference between a discrete choice and its continuous relaxations.

Instead, GCG adopts a greedy search algorithm based on the gradient. In each iteration, it computes $\mathcal{L}(\hat{s}^i)$ for $B$ suffix candidates $\hat{s}^1, \cdots, \hat{s}^B$ and keeps the one with the best loss. The $B$ candidates are obtained by randomly changing one token from the current suffix $s$ and replacing it with a randomly sampled token using the top $K$ tokens. For example, suppose we change the token at position $j$, we first compute the gradient $-\nabla_{e_{s_j}} \mathcal{L}(s)$ with respect to the one-hot vector $e_{s_j}$ and obtain the top $K$ tokens that have the largest gradient. The gradient information is by no means an accurate estimation of the resulting loss because of the gap between the continuous gradient information and the discrete one-hot vector denoting the choice of a token, so we need to check if the resulted new suffix $\hat{s}^i$ leads to a lower loss $\mathcal{L}(\hat{s}^i)$.

To obtain the $B$ candidates, one just needs to perform one forward pass and one backward pass. But to compute the loss for the $B$ candidates, one needs to perform $B$ forward passes. In GCG, $B$ is set to $512$ for optimal performance, making the loss computation the most time-consuming part. As such, we focus on reducing the time cost of the loss computation of the $B$ candidates in this work.

## 2.2 Probe Sampling

**Overview.** As mentioned earlier, the most time consuming part in the GCG algorithm is the loss computation on $B$ suffix candidates $\hat{s}^1, \cdots, \hat{s}^B$. As shown in speculative sampling (Chen et al., 2023; Leviathan et al., 2023), the speculated results using a smaller draft model can be helpful in reducing the computation with a large target model. The original speculative sampling is created to accelerate decoding so it isn't directly applicable here. But the intuition of relying a weaker draft model is obviously useful for negative log likelihood loss computation. Applying the intuition to the problem at hand, we can filter out the suffix candidates that the draft model finds to be unpromising, since the goal is to find the candidate that has the lowest loss with the target model.

In addition, a unique structure in the GCG algorithm is that all the suffix candidates are based on changing one token of the original suffix $s$. As a result of this locality property, it is not unreasonable to assume that one can determine how much they agree on the $B$ candidates based on their agreement on a subset of the $B$ candidates. If the two models agree, we can choose to safely rely on the draft model and filter out more candidates.

Based on these intuitions, we design the `Probe sampling` algorithm as follows: (i) probe agreement between the target model and the draft model to determine the size of the filtered set; (ii) rank candidates using the draft model and obtain the filtered set; (iii) pick the best candidate from the filtered set using the target model.

**Algorithm description.** For the first step, specifically, we sample a probe set comprised of $k$ candidates $\bar{s}^1, \cdots, \bar{s}^k$ and compute their losses using the draft model and the target model and obtain $\mathcal{L}_{\text{draft}}(\bar{s}^1), \cdots, \mathcal{L}_{\text{draft}}(\bar{s}^k)$ and $\mathcal{L}_{\text{target}}(\bar{s}^1), \cdots, \mathcal{L}_{\text{target}}(\bar{s}^k)$. Then we measure the probe agreement score as the Spearman's rank correlation coefficient (Zar, 2005) between the two results as the agreement score. The probe agreement score $\alpha$ is computed as

$$\alpha = 1 - \frac{3 \sum_{i=1}^{k} d_i^2}{k(k^2 - 1)}, \tag{2}$$

where $d_i$ is the distance between the ranks of suffix $\bar{s}^i$ in the two results. For example, $d_i = 4$ if the suffix $\bar{s}^i$ is ranked as number 6 and number 2 for its losses computed from the draft model and the target model. The agreement score $\alpha$ falls into $[0, 1]$ with 1 meaning a full agreement and 0 indicating a non-agreement. We use the rank agreement because it is more robust to the specific values of the resulting loss when measured on drastically different LLMs.

**Algorithm 1** Probe Sampling

---

**Input:** Original suffix $s$, a batch of suffix candidates $\{\hat{s}^1, \cdots, \hat{s}^B\}$, loss function using the draft model and the target model $\mathcal{L}_{\text{draft}}(\cdot)$, $\mathcal{L}_{\text{target}}(\cdot)$.

1: ***Parallel Begin***
2: //Compute loss of all candidates using the draft model
3: **for** $\hat{s}^i \in \{\hat{s}^1, \cdots, \hat{s}^B\}$ **do**
4:     Compute $\mathcal{L}_{\text{draft}}(\hat{s}^i)$
5: **end for**
6: //Compute loss of the probe set on target model
7: $\{\bar{s}^1, \cdots, \bar{s}^k\} = Uniform(\{\hat{s}^1, \cdots, \hat{s}^B\}, k)$
8: **for** $\bar{s}^i \in \{\bar{s}^1, \cdots, \bar{s}^k\}$ **do**
9:     Compute $\mathcal{L}_{\text{target}}(\bar{s}^i)$
10: **end for**
11: ***Parallel End***
12: //Calculate agreement score
13: $\alpha = \text{Spearman\_Cor}(\{\mathcal{L}_{\text{target}}(\bar{s}^i)\}, \{\mathcal{L}_{\text{draft}}(\bar{s}^i)\})$
14: //Evaluate using the target model
15: filtered\_set $= \text{argmin}_{\max\{1, (1-\alpha)B/R\}}\mathcal{L}_{\text{draft}}(\hat{s}^i)$
16: **for** $\hat{s}^i \in$ filtered\_set **do**
17:     Compute $\mathcal{L}_{\text{target}}(\hat{s}^i)$
18: **end for**
19: Output the best suffix in the probe set and the filtered set
20: $s' = \text{argmin}\{\mathcal{L}_{\text{target}}(\bar{s}^i), \mathcal{L}_{\text{target}}(\hat{s}^i)\}$
**Output:** $s'$

---

After obtaining the agreement score, we keep $(1 - \alpha) * B/R$ candidates where $(1 - \alpha) * B$ means that the filtered set size is a scale-down of the previous batch size $B$ and $R$ is a hyperparameter that determines a further scale down. When $\alpha$ is close to $0$, meaning little agreement between the two models, we will use a filtered set size of $B/R$. When $\alpha$ goes to $1$, we almost filter out most of the candidates. With the filtered size determined, we can readily rank the candidates according to the draft model and filter the ones with higher losses. Finally, we evaluate the final loss on the filtered set using the target model and select the best candidate.

**Details.** At first glance, probe sampling involves extra computation but it actually achieves effective acceleration. For computing the losses on the probe set using both the draft model and the target model, the size of the probe set can be set to be relatively small, so it would not add too much to the total time cost. The ranking procedure involves sorting on CPU, but luckily the probe set is small enough that this doesn't become a bottleneck. And the loss computation using the draft model on the whole candidate set is relatively cheap because of draft model's small size. These two operations can also be parallelized on GPU. On the plus side, we are able to avoid computing the loss using the big target model on many candidates that are filtered out. As we will show in the experiments, this approach achieves significant speedup measured by both running time and #FLOPs.

An alternative to computing agreement on the spot is to measure the agreement score on a predetermined set of candidates and use a fixed agreement score for all the suffixes. This would save the time used to measure agreement for each candidate set. However, as we will show in the experiment, this approach does not work so well in terms of speedup. Our intuition is that one can squeeze the time cost more effectively if the agreement is measured accurately, and an adaptive agreement score is more accurate than an one-size-fits-all score. The plausibility of the adaptive score comes back to the locality property that we discussed earlier. Given a specific candidate set, one can accurately estimate the agreement because all the suffixes in this candidate set are similar to a large extent. However, given another candidate set altered from a different suffix, the agreement of the draft model and the target model can be widely different.

In practice, we adopted two small changes in our implementation. First, we do not have a separate step to compute the loss of the probe set candidates using the draft model, since we need to compute the loss on all candidates for filtering purposes. We simply get the numbers from the losses on the whole candidate set. Second, to get the best candidate for the final result, we also look at the losses on the probe set, since the target model is evaluated on the probe set. Ideally, the candidates in the

probe set should be in the filtered set if they achieve a low loss. However, it also does not hurt to look at the best candidate in the probe set in case it is not included in the filtered set. The overall algorithm is further illustrated in Algorithm 1, and the corresponding implementation is shown in Appendix A. We also test simulated annealing (Pincus, 1970) that provides complementary benefit to our algorithm.

## 2.3 Applying Probe Sampling to Other Prompt Optimization Methods

Although prompt sampling was designed to accelerate GCG, the general idea of reducing forward computation can be applied on other prompt optimization methods, where there is usually a process of sampling prompt candidates and evaluating their performances. To see whether probe sampling can effectively accelerate other methods, we also apply probe sampling to two prompt learning methods AutoPrompt (Shin et al., 2020) and APE (Zhou et al., 2022). In addition, we apply probe sampling on AutoDAN (Liu et al., 2024), a genetic algorithm that can find natural jailbreak prompts.

# 3 Experiment

In this section, we evaluate the proposed method on its efficacy and the important factors through extensive studies.

## 3.1 Experiment Details

**Settings.** Following the original GCG paper, we conduct experiments on the first 100 instances of AdvBench (Zou et al., 2023), which are divided into two parts, 500 harmful strings and 500 harmful human behaviors. We test open-source LLMs that have been specifically fine-tuned with respect to safety, including Llama2-chat-7b (Touvron et al., 2023) and Vicuna-7b (Zheng et al., 2023). In the case of draft models, in our main experiments, we use a much smaller model GPT-2 (Radford et al., 2019). Similarly, when applying probe sampling to AutoDAN, we use Llama2-7b-chat as the target model and GPT-2 as the draft model. For AutoPrompt, we follow their original setting, which uses RoBERTa-large (Liu et al., 2019) as the target model and tests on SST-2 (Socher et al., 2013) for sentiment analysis and SICK-E (Marelli et al., 2014) for natural language inference. We use the RoBERTa-base model as the draft model. In the case of APE, we conduct the experiments on three widely used benchmarks GSM8K (Cobbe et al., 2021), BBH (Suzgun et al., 2023), and MMLU (Hendrycks et al., 2020). For these experiments, we use Vicuna-7b-v1.5 as the target model and GPT-2 as the draft model.

**Evaluation.** Following (Zou et al., 2023), we use Attack Success Rate (ASR) as the evaluation metric for GCG and AutoDAN, which is defined as the percentage of inputs that successfully lead LLMs to generate harmful outputs. An output is determined to be harmful if it does not match with rejection phrases, such as "I'm sorry", "I apologize" and "As an". This is not a perfect measurement but works relatively well in practice since LLMs are trained to reject harmful replies. It is also easy to measure and interpret. For prompt learning methods tested on other tasks, we employ Accuracy (Acc) as the metric. The processing time is determined as the average time used for each iteration across all input samples and all iterations. In all experiments, we use 1 NVIDIA A100 GPU with 80GB memory unless mentioned otherwise.

**Hyperparameters.** To determine the hyperparameters for probe sampling, including probe set size $k$, filtered set size reduction hyperparameter $R$, we construct a validation set of size 100 from AdvBench by random sampling in the 400 instances different from the test set. We follow (Zou et al., 2023) for the hyperparameters used in the original algorithm such as the size of the candidate set $B$. We provide detailed analysis of hyperparameters in Section 3.4. When we combine probe sampling with simulated annealing, we follow the same procedure to select hyperparameters. We use the same number of optimization steps 500 as in GCG throughout the paper.

## 3.2 Main Results

**Acceleration results.** As shown in Table 1, probe sampling achieves a speedup of 5.6 times and 6.3 times on Human Behaviors and Human Strings with Llama2 when combined with simulated

Table 1: Comparing the ASR and processing time of `Probe sampling` with and without simulated annealing to GCG with and without simulated annealing, while measuring time and FLOPs by averaging each iteration.

| Model | Method | Harmful Strings | | | Harmful Behaviors | | | | |
| | | ASR | Time (s) | #FLOPs | Individual ASR | Multiple ASR (train) | ASR (test) | Time (s) | #FLOPs |
|---|---|---|---|---|---|---|---|---|---|
| Vicuna (7b-v1.3) | GCG | 88.0 | 4.1 | 97.3 T | 99.0 | **100.0** | 98.0 | 4.8 | 106.8 T |
| | GCG + Annealing | 89.0 | 1.5 (2.7×) | 38.5 T | 98.0 | 92.0 | 94.0 | 2.1 (2.3×) | 46.2 T |
| | `Probe sampling` | 91.0 | 1.7 (2.4×) | 42.4 T | **100.0** | 96.0 | 98.0 | 2.3 (2.1×) | 53.2 T |
| | PS + Annealing | **93.0** | **1.1 (3.6×)** | **27.8 T** | **100.0** | 96.0 | **99.0** | **1.5 (3.2×)** | **24.7 T** |
| Llama2 (7b-Chat) | GCG | 57.0 | 8.9 | 198.4 T | 69.0 | 88.0 | 84.0 | 9.2 | 202.3 T |
| | GCG + Annealing | 55.0 | 2.4 (3.9×) | 39.7 T | 68.0 | 92.0 | 88.0 | 2.7 (3.4×) | 50.6 T |
| | `Probe sampling` | **69.0** | 2.2 (4.1×) | 43.8 T | **81.0** | 92.0 | **93.0** | 2.6 (3.5×) | 40.7 T |
| | PS + Annealing | 64.0 | **1.4 (6.3×)** | **31.2 T** | 74.0 | **96.0** | 91.0 | **1.6 (5.6×)** | **32.3 T** |

Table 2: Transferability of `Probe sampling` with different draft models.

| Method | Direct | Transfer | |
| | Llama2-7b | Vicuna-7b | Mistral-7b |
|---|---|---|---|
| GCG | 69.0 | 89.0 | **86.0** |
| PS (GPT-2) | 85.0 | 92.0 | 83.0 |
| PS (ShearedLlaMa) | **91.0** | **93.0** | 85.0 |
| PS (Flan-T5) | 57.0 | 78.0 | 69.0 |

Table 3: Transferability of `Probe sampling` with different filtered set size $(1 - \alpha) * B/R$.

| Method | Direct | Transfer | |
| | Llama2-7b | Vicuna-7b | Mistral-7b |
|---|---|---|---|
| GCG | 69.0 | 89.0 | **86.0** |
| PS ($R = 64$) | 60.0 | 77.0 | 74.0 |
| PS ($R = 8$) | **85.0** | **92.0** | 83.0 |
| PS ($R = 1$) | 79.0 | 88.0 | 84.0 |

annealing. Probe sampling achieves a speedup of $3.5$ and $4.1$ times alone. With Vicuna, we achieve an overall speedup of $3.2$ and $3.6$ respectively on the two datasets. We also measure the #FLOPs for different settings and found that the speedup results reflects in the reduction of #FLOPs. For example, with Llama2, the #FLOPs reduction is $202.3T/32.3T = 6.3$ times and $198.4T/31.2T = 6.4$ times on the two sets, which is close to the actual speedup results. This also shows that our algorithm results in little overhead with the introduced new procedures. It is worth noting that simulated annealing also achieves decent acceleration and is complementary to our acceleration results.

**GCG results.** Interestingly, we achieve a better ASR score than the GCG algorithm although technically acceleration introduces noise to the algorithm. For instance, with Llama2, we improve the ASR from $57.0$ to $64.0$ on Human Strings and from $84.0$ to $91.0$ on Human Behaviors. We hypothesize that the improvement comes from the randomness added to the GCG algorithm based on greedy search over a single objective. Introducing randomness and noise has been seen as one of the advantages of SGD over full batch training. In contrast, simulated annealing only leads to comparable ASR when applied on GCG.

**Transferability** Table 2 shows probe sampling's transferability across draft models based on Llama2-7b-Chat to various target models. We find that it maintains transferability when using draft models like GPT-2 and SheardLlaMa, which preserve the original ASR of plain GCG. However, draft models that significantly degrade initial performance, such as Flan-T5, impair transferability. Table 3 examines probe sampling transferability across filtered set sizes. Results align with prior findings: probe sampling minimally impacts transferability with appropriate parameters but decreases performance when Llama2-7b-chat's direct ASR is low such as $R = 64$.

**Results on AutoDAN, Autoprompt and APE.** Table 5 demonstrates the effective acceleration of AutoPrompt through the implementation of probe sampling, resulting in a speedup of $1.79\times$ on SST-2 and $1.83\times$ on SICK-E. Importantly, this acceleration is achieved without compromising performance, as evidenced by the minimal changes in accuracy from $91.4$ to $90.6$ on SST-2 and from $69.3$ to $68.9$ on SICK-E. Furthermore, the application of probe sampling to APE, as presented in Table 6, results in significant speed improvements, with a speedup of $2.3\times$ on GSM8K, $1.8\times$ on MMLU, and $3.0\times$ on BBH. Similarly, these speed enhancements do not compromise the performance of APE. In addition, we implement probe sampling on another jailbreak method, AutoDAN. The detailed results can be found in Table 4. Our findings indicate that probe sampling can achieve a speedup of $2.3\times$ for AutoDAN-GA and $2.5\times$ for AutoDAN-HGA, while minimally affecting its performance.

Table 4: Performance of `Probe sampling` on accelerating AutoDAN.

| Method | ASR | Time (s) |
|---|---|---|
| AutoDAN-GA | **56.2** | 424.2 |
| AutoDAN-GA + PS | 55.9 | **182.7 (2.3×)** |
| AutoDAN-HGA | 60.8 | 237.9 |
| AutoDAN-HGA + PS | **62.1** | **95.3 (2.5×)** |

Table 5: Performance of `Probe sampling` on accelerating prompt learning method AutoPrompt.

| Method | SST-2 | | SICK-E | |
|---|---|---|---|---|
| | Acc | Time (s) | Acc | Time (s) |
| Original | 85.2 | N / A | 49.4 | N / A |
| Autoprompt | **91.4** | 228.4 | **69.3** | 42.7 |
| Autoprompt + PS | 90.6 | **127.2 (1.8×)** | 68.9 | **23.6 (1.8×)** |

Table 6: Performance of `Probe sampling` on accelerating prompt learning method APE.

| Method | GSM8K | | MMLU | | BBH | |
|---|---|---|---|---|---|---|
| | Acc | Time (s) | Acc | Time (s) | Acc | Time (s) |
| Vicuna | 20.4 | N/A | 45.6 | N / A | 38.6 | N / A |
| APE | 21.3 | 431.8 | **48.2** | 187.3 | **40.8** | 265.2 |
| APE+PS | **22.4** | **192.3 (2.3×)** | 47.3 | **102.5 (1.8×)** | 39.9 | **88.7 (3.0×)** |

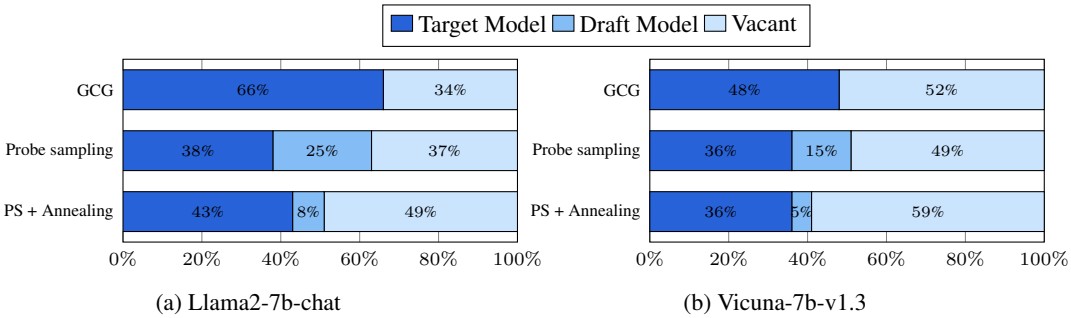

(a) Llama2-7b-chat                     (b) Vicuna-7b-v1.3

Figure 3: Memory usage on a single A100 with 80GB memory with (a) Llama2-7b-chat and (b) Vicuna-7b-v1.3 on 1 instance. The memory consumption of probe sampling with or without simulated annealing is similar to that of the original setting. The computation with the target model still takes most of the memory.

These results demonstrate the effectiveness of our method in not only accelerating GCG but also its applicability to general prompt optimization methods and other LLM attack methods.

### 3.3 Computation Detail Analysis

**Memory allocation.** We evaluate whether probe sampling uses more memory because of the use of an extra model. In Figure 3, we show the memory usage of GCG, probe sampling with and without annealing using either Llama2-7b-chat and Vicuna-7b-v1.3. Probe sampling uses a similar amount of memory to the original GCG algorithm although it involves extra procedures and an extra model, by saving the computation of target model on the whole candidate set. As such, the usage of probe sampling does not introduce extra memory and can be applied when the original GCG algorithm is applied. In terms of the memory usage of the target model and the draft model, most of the memory is spent on target model, probably because the draft model is much smaller.

**Time allocation.** We look at the specific time spent on different operations. As shown in Figure 4, probe set computation using the target model and full set computation using the draft model take a similar amount of time so we can parallelize the computation easily. Sampling candidates in the graph involves a forward and backward pass as mentioned earlier and can be completed relatively quickly. Similarly, it is also fast to compute the agreement using the ranked losses on CPU, so our algorithm introduces relatively little overhead.

### 3.4 Further analysis

In this section, we conduct extensive studies to understand how the proposed method works. We conduct all of the following experiments on the validation set, so the numbers are not directly

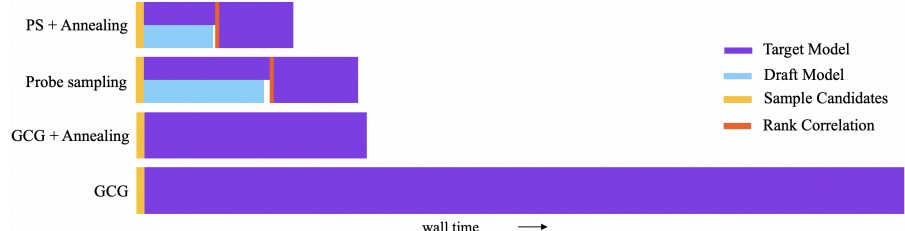

Figure 4: Wall time of GCG, probe sampling with and without simulated annealing. For the target model computation, the first part is done on the probe set and the second part is done on the filtered set. Draft model computation and computation of the target model on the probe set are suited to be done in parallel as they take similar time.

Table 7: Ablation on the filtered set size reduction $R$. The filter set size is $(1 - \alpha) * B/R$.

| Reduction $R$ | 64 | 16 | 8 | 4 | 2 | 1 |
|---|---|---|---|---|---|---|
| ASR | 60.0 | 70.0 | **85.0** | 81.0 | 76.0 | 79.0 |
| Time (s) | **2.01** | 2.31 | 2.60 | 3.02 | 3.41 | 5.19 |

Table 8: Ablation on fixed probe agreement score $\alpha$ vs adaptive score.

| Agreement $\alpha$ | 0.9 | 0.6 | 0.3 | 0.0 | Adaptive |
|---|---|---|---|---|---|
| ASR | 70.0 | 77.0 | 75.0 | 81.0 | **85.0** |
| Time (s) | **2.17** | 2.41 | 2.71 | 3.01 | 2.60 |

comparable to the numbers in the main results. For the validation set, the original GCG algorithm achieves an ASR of 66.0 with an average time of 9.16 seconds per iteration. In each of the study, we highlight the settings that we find to be the best.

**Filtered set size.** The filtered set size is the most important factor in our method. If it is too small, then we will achieve a lot of speedup at the cost of relying too heavily on the draft model and resulting in a lower ASR. If it is too big, then we would not achieve much speedup. Hence we experiment with different filtered size reduction hyperparameter $R$. The filter set size is $(1 - \alpha) * B/R$ where $\alpha$ is the probe agreement score described in Section 2.2.

As shown in Table 7, the time does monotonically decrease if we use a smaller filtered set size. However, interestingly, there is a sweetspot for the ASR with $R$ set to 8. We believe that this can resonates with the hypothesis of introducing randomness as the source of ASR boosts. Both too much or too little randomness hurt performance. As such, we use $R = 8$ for probe sampling. We further show several convergence processes with varying values of $R$ in Appendix B.

**Adaptive vs fixed filtered set size.** As mentioned in Section 2.2, an alternative to use an adaptive filtered set size is to use a fixed size. Here we investigate whether it matters to use an adaptive filtered set size that is determined by how much the draft model and the target model agree on each candidate set. To use a fixed size, we simply fix the probe agreement score $\alpha$ to be 0.9, 0.6, 0.3, and 0.0 and compare with the adaptive case. As shown in Table 8, fixed probe agreement scores always lead to worse ASR. Furthermore, when adopting GPT-2 as the draft model, the average agreement score is 0.45 with a standard deviation of 0.11. This shows that the agreement score between the two models varies significantly for different candidate sets. We also provide the statistics of $\alpha$ for other draft models in Table 11.

**Probe agreement measurement.** We also experiment alternatives to measure the probe agreement score, including the Pearson correlation coefficient (Pearson, 1900), Kendall's Tau correlation coefficient (Kendall, 1938), and Goodman and Kruskal's gamma (Goodman et al., 1979) where the Pearson correlation coefficient directly uses the loss values to compute the agreement and the others use the ranking information. As shown in Table 9, all methods have similar time cost, and Spearman's rank correlation coefficient achieves the best ASR. The Pearson correlation coefficient performs worse than other ranking-based agreement measurement.

**Probe set size.** The size of the probe set also determines whether the probe agreement score is measured accurately. As such, we experiment with different probe set size and report the performance in Table 10. We find that using a small probe set such as $B/64$ or $B/32$ can result in inaccurate

Table 9: Ablation on probe agreement measurements. All methods achieve similar speedup while Spearman's rank correlation coefficient achieves the best ASR.

| Cor | Spearman | Pearson | Kendall | Kruskal |
|---|---|---|---|---|
| ASR | **85.0** | 70.0 | 74.0 | 79.0 |
| Time (s) | 2.60 | 2.47 | 2.53 | **2.43** |

Table 10: Ablation on the probe set size $k$. Using $B/16$ leads to accurate probe agreement measurement while achieving significant acceleration.

| Probe | $B/64$ | $B/32$ | $B/16$ | $B/4$ | $B/2$ | $B$ |
|---|---|---|---|---|---|---|
| ASR | 64.0 | 72.0 | 85.0 | 86.0 | 85.0 | **87.0** |
| Time (s) | **2.10** | 2.57 | 2.60 | 3.41 | 5.61 | 9.58 |

Table 11: Experiments with different draft models. Models with over 1B parameters, like TinyLlama, Phi, and SharedLlMa, need two GPUs for parallel computation. ShearedLlMa achieves the highest ASR probably because it is a pruned version of Llama2. Both GPT-2 and GPT-Neo achieve a good balance of ASR and speedup.

| Model | 1 GPU | | | | 2 GPUs | | |
| | GPT-2 (124M) | GPT-Neo (125M) | Flan-T5 (248M) | BART (406M) | TinyLlama (1.1B) | Phi (1.3B) | ShearedLlaMa (1.3B) |
|---|---|---|---|---|---|---|---|
| $\alpha$ | $0.45 \pm 0.10$ | $0.51 \pm 0.11$ | $0.61 \pm 0.13$ | $0.46 \pm 0.09$ | $0.52 \pm 0.13$ | $0.52 \pm 0.11$ | $\mathbf{0.35} \pm 0.12$ |
| ASR | 85.0 | 81.0 | 57.0 | 76.0 | 72.0 | 82.0 | **91.0** |
| Time (s) | **2.60** | 2.82 | 3.89 | 2.93 | 3.38 | 4.83 | 3.93 |

agreement score, which put a put a significant toll on the attack success rate. It also does not lead to too much time reduction since the draft model computation done in parallel takes more time and the reduced computation is not the bottleneck. Using a larger probe set size such as $B/4$ and $B/2$ will lead to more accurate agreement score but does not increase the ASR significantly. As such, using a probe set of size $B/16$ is good enough to accurately measure the agreement and achieves maximum time reduction.

**Draft model study.** Here we also experiment with bigger draft models, some of which is of similar size to Llama2. We experiment with GPT-Neo (Gao et al., 2020), Flan-T5-base (Chung et al., 2024), BART (Lewis et al., 2019), Phi-1.5 (Li et al., 2023), TinyLlama (Zhang et al., 2024) and Sheared-LLaMA (Xia et al., 2023). Among them, Sheared-LLaMA might be the closest to Llama2 since it is a pruned version of Llama2. For TinyLlama, Phi and Sheared-LLaMA, we use 2 A100s with 80GB memory to fit the whole computation.

As shown in Table 11, Sheared-LlaMa achieves the best ASR although the time reduction is not as good as smaller models such as GPT-2 and there would be a higher time cost if we manage to fit all computation in one GPU. On contrast, Flan-T5, BART, TinyLlama and Mistral all achieve lower ASRs probably because of being very different than Llama2. However, the results are still better than the baseline ASR 66.0. GPT-2 and GPT-Neo achieve a good balance of performance and speedup.

# 4 Related Work

**Alignment of LLMs.** To build safe LLMs, alignments has also been a widely studied topic in the community (Stiennon et al., 2020; Ouyang et al., 2022). Efforts have been put into improving helpfulness (Bai et al., 2022a; Cheng et al., 2023), honesty (Kaddour et al., 2023; Liu et al., 2023; Xu et al., 2023), and harmlessness (Hartvigsen et al., 2022). Among these works, there has been a growing interest in using feedback from a LLM to perform alignment (Bai et al., 2022b; Gulcehre et al., 2023; Burns et al., 2024; Yuan et al., 2024a). Despite all the efforts, there has not been a definitive answer for LLM safety alignments, which also motivates our research in LLM safety.

**Discrete Prompt Optimization.** Attacking LLMs via adversarial prompt can be formulated as a discrete prompt optimization problem (Zou et al., 2023). In this context, attacking algorithms strive to discover superior prompts that effectively steer aligned LLMs toward generating adversarial answers. Some approaches leverage LLMs themselves to iteratively refine prompts (Xu et al., 2022; Pryzant et al., 2023). However, aligned LLMs may resist refining adversarial prompts, rendering these methods ineffective. Other strategies employ RL-based prompt optimization techniques such as those in (Mingkai and Jianyu, 2022; Lu et al., 2023), necessitating additional MLP training with

extensive adversarial data and specific reward design. Moreover, other models introduced in (Cho et al., 2023; Long et al., 2024) to help with prompt optimization must remain unaligned, particularly in jailbreak scenarios (Chao et al.). However, their performance tends to be limited, especially when dealing with strongly fine-tuned models like Llama2-Chat.

**LLM Jailbreaks.** LLM Jailbreaks have received considerable interests recently since due to the implications of applying LLMs widely in human society. Although there is a continuous effort to build safe and reliable LLMs, bypassing the safety mechanism of LLMs is not uncommon. For example, fine-tuning a safe LLM on a few data instances can easily breaks its safety guarantees (Qi et al., 2024; Lermen and Rogers-Smith, 2024). Treating the jailbreak as a prompt optimization problem has also led to a certain level of success (Zou et al., 2023; Mökander et al., 2023; Liu et al., 2024; Chao et al.; Geisler et al., 2024). In addition, conversing in a ciphered language (Yuan et al., 2024b), planting a backdoor during RLHF (Rando and Tramèr, 2023), using a less well-aligned language (Deng et al., 2024) and multi-modality (Shayegani et al., 2024) can also lead to successful jailbreaks. Researchers also construct large dataset of manual jailbreak prompts (Toyer et al., 2023).

Among these jailbreak methods, the prompt optimization method GCG (Zou et al., 2023) provides the more general and universal solution for us to study the jailbreaking problem. As such, in this work, we mainly focus on the acceleration of GCG, but the idea of delegating computation to a draft model can also be applied in other situations such as the multi-modality case and finetuning case. We leave the extension of this work for future work.

**Acceleration.** In the field of acceleration, speculative sampling (Chen et al., 2023; Leviathan et al., 2023) is the most relevant to our method. They also use a draft model but its design cannot be directly applied to accelerate the GCG algorithm. REST (He et al., 2024) adopts the concept of speculative sampling but uses a retrieval approach based on a Trie to construct the candidate. The attention module has also been a focus of acceleration because of its quadratic nature (Dao et al., 2022; Cai et al., 2024). There have also been continuous interests in more efficient versions of Transformers (So et al., 2019; Dai et al., 2021; Liu et al., 2021; Gu et al., 2020, 2021). These architectural changes are complementary to our algorithm design and we leave it to future work.

## 5 Conclusion

In this paper, we propose an algorithm probe sampling that can effectively accelerate the GCG algorithm. We achieve an acceleration ranging from $2.1\times$ to $6.3\times$ in different scenarios on AdvBench. We illustrate the intuition and how the algorithm works through extensive experiments. Furthermore, this approach is also applied to general prompt optimization methods and other jailbreak techniques, including AutoPrompt, APE, and AutoDAN. We believe the idea of using the probe agreement score to perform adaptive computation can be applied to cases other than GCG. For example, it could potentially be used to perform conditional computation for attention. Another direction is to extend the framework to the multi-modality case which can be interesting given the vast amount of video data. It would also be interesting to run a small draft model on the scale of web data to detect the existence of natural adversarial prompts.

## Limitation and Impact Statements

Probe sampling has two main limitations. Firstly, it exhibits relatively slow performance when tested on large-sized test sets, which hampers its efficiency. Secondly, it is limited to supporting only open-source models, thereby excluding proprietary or closed-source models from benefiting from the proposed acceleration techniques. These limitations indicate the need for further improvements to enhance the speed and broaden the model support in order to make the jailbreak acceleration approach more robust and applicable across a wider range of language models.

Probe sampling can be applied to accelerate GCG algorithm. Having a faster algorithm to explore adversarial cases of alignments enable us to study how to make LLMs safer. As far as we know, as of now, there is not a LLM that can use this algorithm to achieve malicious behavior in real-world that would not be possible without the algorithm. The goal of this research is to present a general algorithm which may inspire new research, and also contribute to the gradual progress of building safe and aligned AIs.

## Acknowledgements

This research is partially supported by the National Research Foundation Singapore under the AI Singapore Programme (AISG Award No: AISG2-TC-2023-010-SGIL) and the Singapore Ministry of Education Academic Research Fund Tier 1 (Award No: T1 251RES2207). Xuan Long Do is supported by the A*STAR Computing and Information Science (ACIS) scholarship. We thank Liwei Kang for insightful discussion, Liying Cheng for helping with plotting figures.

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

# A   Implementation

The following code shows the core implementation of probe sampling using PyTorch. As seen in the code, the algorithm is relatively easy to use.

```python
def draft_model_all(args):
    draft_model.loss(control_cands)

    queue.put('draft':loss_small)

def target_model_probe(args):
    probe_index = random.sample(range(512), 512/16)
    probe_control_cands = control_cands[probe_index]
    target_model.loss(probe_control_cands)

    queue.put('target':[loss_large_probe, probe_index])

# Parallelly Calculate Loss on Batch and Probe Set
args=(control_cands, batch_size, queue)
threading.Thread(target=draft_model_all, args=args)
threading.Thread(target=target_model_probe, args=args)

# Calculate Agreement Score
cor = spearmanr(loss_small[probe_index], large_loss_probe)

# Target Model Test on Filtered Set
filtered_size = int((1 - cor) * 512/8)
indices = topk(loss_small, k=filtered_size, largest=False)
filtered_control_cands = control_cands[indices]
target_model.loss(filtered_control_cands)

# Return Lowest Loss Candidate
return [large_loss_probe, filtered_control_cands].lowest()
```

# B   Converge Process

In Figure 5, we also show a few convergence processes with different values of $R$, where the pink line corresponds to $R = 8$. The pink line always achieves successful optimization while the other lines can lead to suboptimal results due to excessive randomness or insufficient randomness. In particular, the blue and yellow lines can suffer from excessive randomness and the other lines might have insufficient randomness.

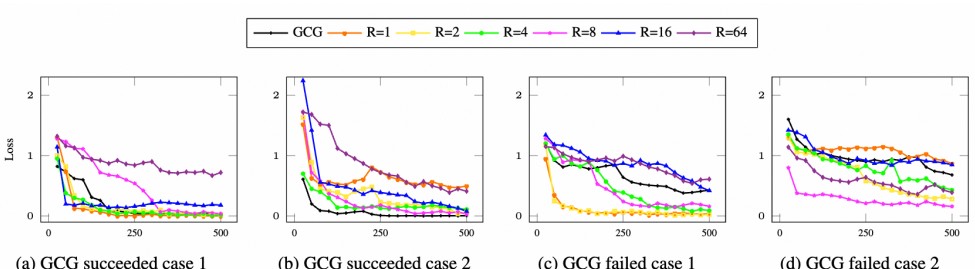

Figure 5: Converge progress with different sizes of filtered set.

# C   Software optimization

In other speedup works (He, 2023), using *torch.compile()* can lead to significant acceleration. It compiles LLMs into an kernel and alleviate the overhead of repeatedly launching the kernel. Table 12 shows that the time cost is similar with or without this optimization enabled. This is likely due to the fact that we use large batch sizes and long input sequences, whose computation cost dominates the overhead caused by the eager execution and launching the kernel repeatedly.

Table 12: Results with torch.compile() enabled. torch.comple() does not lead to further speedup.

| Method | GCG | Probe sampling | PS (Compile) |
|---|---|---|---|
| ASR | 66.0 | 85.0 | 85.0 |
| Time (s) | 9.16 | 2.60 (3.5×) | 2.54 (3.6×) |

