# OpenReview forum: "Accelerating Greedy Coordinate Gradient and General Prompt Optimization via Probe Sampling"
_NeurIPS.cc/2024/Conference — NeurIPS 2024 poster_

### Official Review · Reviewer_hJdQ · 2024-07-04

**Soundness:** 3
**Presentation:** 3
**Contribution:** 2
**Rating:** 8
**Confidence:** 4

**Summary:**

The paper proposes a method to accelerate discrete prompt optimization algorithms. In addition to the target model, a smaller draft model is used to reduce the number of candidates to be evaluated by the target model based on an agreement score between the two models. When applied to Greedy Coordinate Gradient and trained with the open-source LLMs  Llama2-chat-7b and Vicuna-7b, the approach achieves a better runtime while maintaining the attack success rate (ASR) on AdvBench. The method is also applied to other prompt optimization algorithms such as AutoPrompt, ADE and AutoDAN.

**Strengths:**

- The problem studied in the paper is important. The method makes finding prompts for very large models more efficient.
- The method is simple and general enough that it can be applied to any discrete prompt optimization algorithms that rely on LLM candidate evaluation.
- Except for some minor typos, the paper is well-presented and is easy to read.

**Weaknesses:**

- Transferability is a very important aspect of adversarial attacks but it is never mentioned in the paper.
- The related work section lacks a proper presentation of discrete prompt optimization algorithms. Also, the authors did not mention other prior work that also use a pair of models for discrete prompt optimization.

Minor issues:
The use of the term "Probe sampling" is inconsistent across the paper. Sometimes it is not emphasized and in lower case.
line 310: "In this paper, we propose an algorithm called Probe sampling that can ..."

**Questions:**

- Does this method affect the transferability of adversarial attacks to other LMs (for example from Llama2-chat-7b to Vicuna-7b) compared to plain GCG?
- What is the effect of the filtered set size and the draft model (size, similarity to target model) on transferability?
- GCG operates at the token level and the draft model has a different tokenizer. What is the effect of the draft model tokenizer on the resulting suffix? also on transferability? (are tokens common to both tokenizers chosen or is the method completely oblivious to tokenization differences)

**Limitations:**

The authors have included a section on limitations and potential negative social impact.

---

> ### Author Rebuttal · Authors · 2024-08-07
>
> Dear Reviewer hJdQ,
>
> Thank you for your insightful reviews and comments. We appreciate the time and effort you have put into providing valuable feedback. We would like to address your concerns as follows:
>
> > Concern: Transferability of Probe Sampling
>
> We appreciate your concern regarding the transferability of probe sampling, and we conduct corresponding experiments to analyze it.
>
> **1. Whether draft model of probe sampling affect transferability**
>
> We compare probe sampling implemented across various draft models on Llama2-7b-Chat and then transfer to diverse target models. Detailed results are shown as follows. Our investigation reveals that probe sampling has minimal impact on transferability when applied to appropriate draft models that maintain the original ASR of plain GCG. Conversely, utilizing draft models that degrade the initial performance significantly affects transferability.
>
>
> |                   | Llama2-7b-Chat (Direct) | Vicuna-7b (Transfer) | Mistral-7b (Transfer) |
> | ----------------- | :---------------------: | :------------------: | :-------------------: |
> | Plain GCG         |           69            |          89          |        **86**         |
> | PS (GPT-2)        |           85            |          92          |          83           |
> | PS (ShearedLlaMa) |         **91**          |        **93**        |          85           |
> | PS (Flan-T5)      |           57            |          78          |          69           |
>
> In addition to open-source models, we also conduct transfer experiments on GPT4, especially gpt4-0125-preview. Our findings indicate that probe sampling not only markedly accelerates the attacking process but also maintains the transferred ASR. The detailed results are presented below.
>
> |                |    Optimized on    |   ASR    |       Time (s)        |
> | -------------- | :----------------: | :------: | :-------------------: |
> | GCG            | Llama2-7b & Vicuna |   16.0   |          9.8          |
> | Probe Sampling | Llama2-7b & Vicuna | **17.0** | **3.2 (3.1$\times$)** |
>
>
>
> **2. Whether the filtered set size of probe sampling affect transferability?**
>
> In our experiments, we investigate the transferability of probe sampling across various filtered set sizes, denoted as $(1-\alpha)*B/R$. Our results align closely with previous findings, indicating that probe sampling does not significantly impact transferability when appropriate parameters are used. However, it does lead to decreased performance in cases where the direct ASR on Llama2-7b-chat is low.
>
>
> |           | Llama2-7b-Chat (Direct) | Vicuna-7b (Transfer) | Mistral-7b (Transfer) |
> | --------- | :---------------------: | :------------------: | :-------------------: |
> | Plain GCG |           69            |          89          |          **86**           |
> | R=64      |           60            |          77          |          74           |
> | R=8       |          **85**            |          **92**          |          83           |
> | R=1       |           79            |          88          |          84           |
>
> **3. What is the influence of the draft model's tokenizer on suffixes?**
>
> In Table 9, TinyLlama and ShearedLlaMa utilize the same tokenizer as Llama2-7b-Chat. However, their performance diverges significantly. Hence, the tokenizer does not play a pivotal role in effectively attacking the model or the transferability; instead, factors like model performance similarity hold greater importance.
>
> > Suggestion #1: Adding related work
>
> Thank you for recommending more related works, especially in discrete prompt optimization algorithms. We will add the following paragraph in the final version.
>
> **Discrete Prompt Optimization.**  Attacking LLMs via adversarial prompt can be formulated as a discrete prompt optimization problem [1]. In this context, attacking algorithms strive to discover superior prompts that effectively steer aligned LLMs toward generating adversarial answers. Some approaches leverage LLMs themselves to iteratively refine prompts [2, 3]. However, aligned LLMs may resist refining adversarial prompts, rendering these methods ineffective. Other strategies employ RL-based prompt optimization techniques such as those in [4, 5], necessitating additional MLP training with extensive adversarial data and specific reward design. Moreover, other models introduced in [6, 7] to help with prompt optimization must remain unaligned, particularly in jailbreak scenarios[8]. However, their performance tends to be limited, especially when dealing with strongly fine-tuned models like Llama-2.
>
> > Suggestion #2: Align the format of "Probe sampling" in the paper.
>
> We appreciate your effort in seriously looking into the details of our paper. We will address them to enhance our work further in the final version.
>
>
> [1] Zou, et al, Universal and Transferable Adversarial Attacks on Aligned Language Models, Arxiv 2023
>
> [2] Xu, et al, GPS: Genetic Prompt Search for Efficient Few-Shot Learning, EMNLP 2022
>
> [3] Pryzant, et al, Automatic prompt optimization with" gradient descent" and beam search, EMNLP 2023
>
> [4] Deng, et al, RLprompt: Optimizing discrete text prompts with reinforcement learning, EMNLP 2022
>
> [5] Lu, et al, Dynamic prompt learning via policy gradient for semi-structured mathematical reasoning, ICLR 2023
>
> [6] Cho, et al, Discrete Prompt Optimization via Constrained Generation for Zero-shot Re-ranker, ACL 2023
>
> [7] Do, et al, Prompt Optimization via Adversarial In-Context Learning, ACL 2024
>
> [8] Chao, et al, Jailbreaking Black Box Large Language Models in Twenty Queries, Arxiv 2023

---

> ### Comment · Reviewer_hJdQ · 2024-08-08
>
> The authors clearly answered all the questions regarding transferability and the effect of the draft model's tokenizer. The experiments suggest that Probe sampling does not affect the transferability of the adversarial prompts compared to plain GCG. I recommend adding the transferability experiments as well as the discrete prompt optimization algorithms related work in the main paper. I have updated my score.

---

### Official Review · Reviewer_Jngk · 2024-07-11

**Soundness:** 3
**Presentation:** 3
**Contribution:** 2
**Rating:** 4
**Confidence:** 4

**Summary:**

The authors propose using a significantly smaller draft model compared to the target LLM to filter candidate suffixes, thereby accelerating the training process of GCG-based algorithms.
The results demonstrate a faster training speed with enhanced ASR.

**Strengths:**

1. I appreciate the innovative approach of utilizing another LLM to filter suffixes to assist in the attack on the target LLM.
2. The paper is well-organized and written in a formal academic style.

**Weaknesses:**

1. As mentioned on line 82, "one needs to perform B forward passes." In my opinion, these B candidates can be sent to the LLM together within a batch. By fully utilizing the GPU's parallel technique, it actually does not need to take B times the amount of time. Can an LLM with a smaller batch size achieve faster speeds? If not, I think the author should illustrate why the proposed method can speed up.
2. Since the draft model is not optimized, how to guarantee that the draft model and the target model align well, i.e., $\alpha$ is significantly larger than $0$? In my opinion, GCG will generate meaningless words, and the function of the draft model seems to be to check the meaning of these suffixes and discard  the meaningless ones. I encourage the authors to explain the the meaning of alignment between the draft model and the target model.
3. While the GCG is a popular method and the $6x$ speed is impressive, there are another methods like advprompter [1] which has faster speed and higher ASR. I encourage authors to compare the proposed method with more efficient methods instead of GCG.

[1] Paulus, A., Zharmagambetov, A., Guo, C., Amos, B., & Tian, Y. (2024). Advprompter: Fast adaptive adversarial prompting for llms. arXiv preprint arXiv:2404.16873.

**Questions:**

1. The method Amplegcg [2] achieve better performance than GCG, can the proposed method be applied to speed this traning process?

[1] Liao, Z., & Sun, H. (2024). Amplegcg: Learning a universal and transferable generative model of adversarial suffixes for jailbreaking both open and closed llms. arXiv preprint arXiv:2404.07921.

**Limitations:**

none.

---

> ### Author Rebuttal · Authors · 2024-08-07
>
> Dear Reviewer Jngk,
>
>
> We appreciate the time and effort you have put into providing valuable feedback. However, we respectfully believe there might be some misunderstanding regarding our work. We would appreciate the opportunity to clarify a few points and address your concerns as follows:
>
>
> > Misunderstanding #1:  Probe sampling performs B forward passes in parallel within a batch or sequentially?
>
> We understand your concern related to whether probe sampling can accelerate GCG when B candidates are sent to LLMs in a batch. The answer is definitely YES, as we consistently use this approach throughout our paper, similar to the GCG paper. For more details on how this is implemented, please refer to the GCG code, specifically lines 170 to 174 in `llm-attacks/llm_attacks/gcg/gcg_attack.py`. In this context, the term “B forward passes” should be more accurately articulated as “forward computation on B candidates”. This modification ensures a clearer understanding of the procedure, emphasizing the total amount of computation. Moreover, Figure 3 illustrates why probe sampling can accelerate GCG when concurrently operating B candidates in a batch, owing to its reduced memory usage. A more comprehensive analysis is illustrated in the paper, spanning from line 217 to line 224.
>
>
> > Misunderstanding #2: Draft model is used to check whether the meaning of suffixes are meaningful so how to make sure two models are aligned without training?
>
> The draft model is not used to "check the meaning" of suffixes; it solely assists the target model in ranking candidates. In GCG, we want to find the candidate suffix that achieves the lowest loss, i.e., the suffix that is ranked the first if sorted by their losses computed by the target model. Naturally, if the ranked results agree, the rank using the draft model is indicative of the rank using the target model and we can safely rely on the draft model to filter out more unpromising candidates. Conversely, in cases where $\alpha$ equals 0 ($\alpha$ can not be significantly larger than 0 as it is in the range of [0,1] and 1 meaning a full agreement as explained in line 110), we disregard the rank calculated by the draft model and depend exclusively on the target model in this iteration. Note that $\alpha$​ is adaptively calculated in each iteration (explained from line 134 to line 136 in text and line 13 in Algorithm 1), which significantly contributes to the performance of probe sampling. Furthermore, as illustrated in Table 9, probe sampling works consistently well across different models with various average alignment levels.
>
> > Concern #1: Compare with more efficient methods instead of GCG.
>
> We appreciate your concern about comparing probe sampling with other existing acceleration approaches. AdvPrompter [1] involves training the AdvPrompter with the help of both the target model and another base model, necessitating inference of AdvPrompter for each adversarial prompt. These additional computational resources are significant. According to Figure 2 in [1], AdvPrompter achieves an ASR of 23.4 on Llama2-7b-chat, whereas probe sampling achieves 81. Moreover, following the recommendation from Reviewer bSGi, we also compare probe sampling with BEAST [2] and ACG [3]. It shows that probe sampling not only achieves much higher ASR, but also is orthogonal to them and can further accelerate them. Please refer to the rebuttal addressing concern #1 provided to Reviewer bSGi for further details.
>
> > Concern #2: Whether probe sampling can be used to accelerate AmpleGCG.
>
> We acknowledge the concerns regarding the generalizability of probe sampling. AmpleGCG introduces a methodology that involves initially gathering adversarial attacks through overgenerated GCG and subsequently fine-tune the model using these adversarial input-output pairs. Given that AmpleGCG utilizes GCG as the fundamental algorithm for producing and assembling adversarial outputs, probe sampling can effectively accelerate overgenerated GCG. Following the setting of the paper, we implement probe sampling on overgenerated GCG.
>
>
> |                     | ASR      | Time                  |
> | ------------------- | -------- | --------------------- |
> | GCG                 | 20.0     | 4.3                   |
> | Overgenerate        | 76.7    | 4.3                   |
> | Over+probe sampling | **83.3** | **1.8 (2.4$\times$)** |
>
> We further employ generated adversarial input-output pairs to fine-tune Llama2-7B-chat and test on the same released hard testset with 100 queries. Here are the detailed results.
>
> |                          | AmpleGCG | Probe Sampling Training |
> | ------------------------ | -------- | ----------------------- |
> | Group Beam Search (50)   | **83.0** | 82.0                    |
> | Group Beam Search (100)  | 93.0     | **93.0**                |
> | Group Beam Search (200)  | 99.0     | **100.0**               |
> | Group Beam Search (1000) | **100.0**    | **100.0**               |
>
> We find that adversarial input-output pairs generated by probe sampling accelerated overgenerated GCG achieves nearly the same performance, proving the applicability of probe sampling on generating adversarial training data.
>
> In addition, probe sampling can also be implemented on Group Beam Search algorithm. However, we omit additional experiments here as it is the same as accelerating BEAST as mentioned earlier.
>
>
>
> [1]  Paulus, et al, AdvPrompter: Fast Adaptive Adversarial Prompting for LLMs, Arxiv 2024
>
> [2] Sadasivan, Vinu Sankar, et al, Fast Adversarial Attacks on Language Models In One GPU Minute, ICML 2024
>
> [3] Making a SOTA Adversarial Attack on LLMs 38x Faster, Haizelabs Blog Post, 2024
>
> [4] Liao, et al, AmpleGCG: Learning a Universal and Transferable Generative Model of Adversarial Suffixes for Jailbreaking Both Open and Closed LLMs, Arxiv 2024

---

> ### Author Response · Authors · 2024-08-13
>
> Dear Reviewer Jngk,
>
> I hope this message finds you well. The discussion period is ending soon, I am writing to emphasize the importance of your review for our submission. Your score is significantly lower than the other two reviewers, and we believe this discrepancy may indicate a misunderstanding or oversight.
>
> We have addressed all the concerns in our detailed rebuttal and would appreciate your prompt attention to it. A thorough reassessment is crucial to ensure a fair evaluation.
>
> Your expertise is highly valued, and we trust that a reconsidered review will reflect the true merit of our work.
>
> Thank you for your immediate attention to this matter.
>
> Best regards, Authors

---

### Official Review · Reviewer_bSGi · 2024-07-22

**Soundness:** 3
**Presentation:** 3
**Contribution:** 3
**Rating:** 7
**Confidence:** 4

**Summary:**

This paper presents a novel algorithm called "Probe sampling" to accelerate the Greedy Coordinate Gradient (GCG) method for optimizing adversarial prompts against large language models (LLMs). The key idea is to use a smaller "draft" model to filter out unpromising candidate prompts, reducing the number of expensive evaluations needed on the full target LLM. The method dynamically determines how many candidates to filter based on measuring agreement between the draft and target models on a small probe set. Experiments show Probe sampling achieves 3.5-6.3x speedups on benchmark datasets while maintaining or improving attack success rates. The technique is also shown to accelerate other prompt optimization and LLM attack methods.

**Strengths:**

- Overall: The paper looks solid and the 5.6x runtime improvement for GCG looks compelling. Also, it’s nice to see speed-ups for AutoDAN and prompt learning methods like AutoPrompt and APE. The idea of reducing computation by resorting to a smaller draft model - specifically for forward passes that consume the most time - makes a lot of sense.

- Quality: The empirical evaluation is thorough, testing on multiple datasets and model types. The ablation studies and analysis provide good insight into the algorithm's behavior. The speedups achieved are substantial and practically meaningful.

- Clarity: The paper is generally well-written and easy to follow. The algorithm is well explained, and the experimental setup and results are presented in a clear and organized manner.

**Weaknesses:**

- It would be good to comment on the other existing approaches to speed up GCG like Fast Adversarial Attacks on Language Models In One GPU Minute and Making a SOTA Adversarial Attack on LLMs 38x Faster. Since they were out in February/March, probably it’s still fine to treat them as concurrent work, but it would be good to discuss the differences to better contextualize your work.

**Questions:**

No.

**Limitations:**

The authors provide a reasonable discussion of limitations. One additional limitation that would make sense to discuss is the high memory requirement for gradient-based red-teaming methods like GCG, which effectively limits the experiments to smaller models (i.e., only up to 7B parameters).

---

> ### Author Rebuttal · Authors · 2024-08-07
>
> Dear Reviewer bSGi,
>
> Thank you for your insightful reviews and comments. We appreciate the time and effort you have put into providing valuable feedback. We would like to address your concerns as follows:
>
> > Concern #1: Compare with more related works
>
> We appreciate your concern about comparing probe sampling with other existing acceleration approaches. BEAST [1] and ACG [2] both employ beam-search like decoding methods, accelerating GCG by generating more candidates in each iteration, thereby reducing the total number of iterations. However, probe sampling is orthogonal to these methods, capable of integration into the loss calculation stage of each algorithm.
>
> To verify this claim, we implement probe sampling on BEAST, which can accelerate it by 1.4 times on Vicuna-7b and 1.9 times on Llama2-7b-chat, on average per iteration. This acceleration allows probe sampling to execute more iterations within the same GPU time, thereby improving the ASR. The detailed results are presented below, indicating that probe sampling improves the ASR of both models within the same GPU time.
>
>
>
> |                    | **Clean** |           |       Time (s)        |           | **In one GPU minute (%)** |           | **In two GPU minutes (%)** |
> | :----------------: | :-------: | :-------: | :-------------------: | :-------: | :-----------------------: | :-------: | :-----------------------: |
> |                    |           | **BEAST** |  **Probe-Sampling**   | **BEAST** |    **Probe-Sampling**     | **BEAST** |    **Probe-Sampling**     |
> |   **Vicuna-7B**    |   **7**   |    2.4    | **1.7 (1.4$\times$)** |    89     |          **90**           |  **96**   |          **96**           |
> | **Llama2-7B-chat** |     0     |    4.3    | **2.3 (1.9$\times$)** |     9     |          **13**           |    12     |          **16**           |
>
> Furthermore, as illustrated in our paper (line 150 to line 155), probe sampling can be used to accelerate algorithms involving sampling prompt candidates and evaluating their performances, as evidenced by the experiment results in Table 3 and Table 4.
>
>
> In addition to integrating with beam-search acceleration methods, probe sampling yields better ASR results, with probe sampling achieving a score of 81 on Llama2-7b-chat, surpassing the scores of 12 for BEAST and 64 for ACG.
>
> > Concern #2: Limitation
>
> Thanks for recommending adding the limitation about high memory requirement, we will incorporate the following sentence into the final version.
>
> Although the acceleration of probe sampling does not necessitate additional memory, it still faces the high memory demands inherited from GCG, consequently restricting experiments to smaller models. While simulated annealing offers some relief, the situation remains far from adequate.
>
>
>
> [1] Sadasivan, Vinu Sankar, et al, Fast Adversarial Attacks on Language Models In One GPU Minute, ICML 2024
>
> [2] Making a SOTA Adversarial Attack on LLMs 38x Faster, Haizelabs Blog Post, 2024

---

> > ### Comment · Reviewer_bSGi · 2024-08-12
> > **Follow-up comment**
> >
> > Thanks for the further clarifications. They address my (minor) concerns. I increase my score from 6 to 7.

---

### Decision · Program_Chairs · 2024-09-25

**Decision:**

Accept (poster)

**Comment:**

The paper investigates the problem of generating adversarial prompts using Greedy Coordinate Gradient (GCG) and introduces probe sampling with a small draft model to reduce the costs of prompt evaluation. Experiments demonstrate a substantial speedup on benchmarks. The primary concerns raised were the comparison with recent related work and the alignment between the draft and the base models. During the rebuttal, the authors provided additional experimental results that addressed most of these concerns.